# Did mpox knowledge, attitudes and beliefs affect intended behaviour in the general population and men who are gay, bisexual and who have sex with men? An online cross-sectional survey in the UK

Louise E Smith [1] , Henry WW Potts [2] , Julii Brainard [3] , Tom May [4]
Isabel Oliver [5] , Richard Amlôt [5] , Lucy Yardley [4] , G James Rubin[1]

For numbered affiliations see end of article.

**Correspondence to**
Dr Louise E Smith;
louise.e.smith@kcl.ac.uk

## ABSTRACT

**Objectives**  To investigate rates of mpox beliefs, knowledge and intended behaviours in the general population and in gay, bisexual or other men who have sex with men (GBMSM), and factors associated with intended behaviours. To test the impact of motivational messages (vs a factual control) on intended behaviours.

**Design**  Cross-sectional online survey including a nested randomised controlled trial.

**Setting**  Data collected from 5 September 2022 to 6 October 2022.

**Participants**  Participants were aged 18 years or over and lived in the UK (general population). In addition, GBMSM were male, and gay, bisexual or had sex with men. The general population sample was recruited through a market research company. GBMSM were recruited through a market research company, the dating app Grindr and targeted adverts on Meta (Facebook and Instagram).

**Main outcome measures**  Intention to self-isolate, seek medical help, stop all sexual contact, share details of recent sexual contacts and accept vaccination.

**Results**  Sociodemographic characteristics differed by sample. There was no effect of very brief motivational messaging on behavioural intentions. Respondents from Grindr and Meta were more likely to intend to seek help immediately, completely stop sexual behaviour and be vaccinated or intend to be vaccinated, but being less likely to intend to self-isolate (ps<0.001). In the general population sample, intending to carry out protective behaviours was generally associated with being female, older, having less financial hardship, greater worry, higher perceived risk to others and higher perceived susceptibility to and severity of mpox (ps<0.001). There were fewer associations with behaviours in the Grindr sample, possibly due to reduced power.

**Conclusions**  GBMSM were more likely to intend to enact protective behaviours, except for self-isolation. This may reflect targeted public health efforts and engagement with this group. Associations with socioeconomic factors suggest that providing financial support may encourage people to engage with protective behaviours.

## STRENGTHS AND LIMITATIONS OF THIS STUDY

⇒ A strength of this study is that it collected data from four large samples, including the general population and men who are gay, bisexual or have sex with men.

⇒ Data collection occurred over a short period (5 September 2022 to 6 October 2022) during the mpox outbreak.

⇒ Limitations include that responses may have been affected by social desirability or recall bias, although the anonymous nature of the survey should mitigate this somewhat.

⇒ Sociodemographic characteristics differed by sample, with participants recruited from Grindr and Meta being more likely to be working, highly educated, of higher socioeconomic grade, and have less financial hardship.

⇒ We measured behavioural intentions which may differ from actual behaviours, but factors associated with intentions should still be validly identified.

## INTRODUCTION

Mpox (also known as monkeypox) is an orthopox virus that causes fever, headache, exhaustion, swollen glands and aches (joint, muscle, back), followed by a rash with blisters.[1] It spreads from person to person through touching clothing, bedding or towels used by someone with mpox rash, touching mpox skin blisters or scabs, and through the coughs or sneezes of someone with mpox rash (droplet transmission).[1] Since May 2022, there has been a multicountry outbreak of mpox in non-endemic countries.[2] Estimates from the WHO (data up to 9 August 2023) indicate that there have been over 89 000 cases in 113 countries, resulting in 152 deaths.[3] Most cases have been in men who are



gay, bisexual or have sex with men (GBMSM), with close human skin-to skin contact (including sexual) being the primary driver of transmission.[4] The UK is the eighth country most affected by this mpox outbreak with 3771 cases.[3] Within the UK, most cases have been identified in England (with 69% of English cases in London). Almost all (99%) cases were men; English cases had a median age of 36 years.[5] The peak of the epidemic was seen in June and July 2022, with case numbers falling since the end of July.[6]

In the UK, people who thought they might have mpox were asked to call a sexual health clinic, to stay at home (self-isolate) and avoid close contact with other people.[1] Suspected cases were tested for mpox, and confirmed cases were asked to self-isolate for up to 21 days and engage with contact tracing.[7] Cases, their close contacts, and those most likely to be exposed to mpox were advised to be vaccinated with modified vaccinia Ankara vaccine—which offers cross-protection—to reduce transmission and prevent severe illness.[8 9] While similar public health actions are now familiar to the public as a result of the COVID-19 pandemic, research conducted during the pandemic indicates that engagement with uptake of testing and self-isolation was suboptimal in some groups.[10–12]

Various theories have guided research into the psychological factors associated with the uptake of health behaviours. One such theory is the protection motivation theory (PMT), which states that people's intention to carry out a protective behaviour is influenced by their appraisal of the threat (perceived susceptibility to and severity of, eg, mpox) and the coping response (perceived effectiveness of and self-efficacy for, eg, testing, self-isolation, contact tracing, vaccination).[13] During the COVID-19 pandemic, psychological and sociodemographic factors associated with testing and self-isolating included: higher perceived risk of COVID-19, knowledge of transmission modes, higher perceived effectiveness of protective behaviours, believing that your behaviour had an impact on transmission, and belief that others in the same position would also self-isolate, being female and having less financial hardship.[10–12 14] Generally speaking, engagement with protective behaviours was associated with being older, apart from uptake of lateral flow testing, which was higher in younger people.[10 11] Factors associated with COVID-19 vaccine uptake (completed and intended) included perceiving vaccination to be safe and necessary, perceiving COVID-19 to be more severe, and thinking that others would also be vaccinated.[15–18] Perception of side effects is one of the most common reasons given for refusing vaccination.[19] Historically, smallpox vaccines have been associated with severe adverse effects.[8] While vaccines currently licensed are associated with fewer severe adverse effects, this may affect people's intention to be vaccinated.

At the time of writing, few scientific studies have investigated behaviour during the 2022 mpox epidemic. While most have investigated vaccination acceptability, few have investigated engagement with a contact tracing system. Those studies that have been done suggest that knowledge of transmission modes is incomplete.[20–22] Greater agreement with vaccination for mpox was associated with perceiving the virus as more dangerous and virulent and higher worry about mpox in a survey of members of the Saudi Arabian general public.[23] In Dutch GBMSM, willingness to accept a vaccine was associated with being more worried about getting mpox, perceiving a higher risk of mpox, perceiving mpox to be more severe, thinking that vaccination was important, thinking that the vaccine was effective and greater social norms for vaccination.[24] Another study also conducted in the Netherlands found that vaccination intention in GBMSM was associated with higher worry about mpox, knowing someone who had mpox and being single but dating or in an open/polyamorous relationship.[25] This study also investigated self-isolation intention for 21 days, finding that higher intentions were associated with thinking that mpox had more problematic consequences and lower education.[25] A study conducted in the UK found that agreement with self-isolation was associated with not having completed a higher education degree, not being employed and identifying as having a disability.[26] There was no difference in agreement between GBMSM and those who were not GBMSM. Further research is needed to investigate how psychological factors may affect engagement with public health measures (isolating, testing, contact tracing, vaccination) put in place to control the spread of the mpox outbreak in the UK.

Official communications are vital during new and emerging outbreaks, and serve to inform the public about the threat, the public health response, and behaviours that people should engage with in order to protect themselves and others.[27] Messages based on theories of health behaviours, such as the PMT,[13] may therefore increase engagement with protective behaviours. For example, research suggests that COVID-19 vaccination intention increased when communications emphasised the safety and effectiveness of the vaccine and used social norms interventions (eg, asking people to 'join the millions' being vaccinated).[28] While findings relating to messages emphasising the benefits of vaccination to oneself and others were mixed, there is some evidence that the influence of these messages may be most evident in strongly hesitant groups.

Decreasing rates of incidence of HIV in England[29] suggest that efforts to promote safer sexual practices in GBMSM have been successful. Existing communication channels and protocols to prevent HIV risk may have led to increased knowledge and awareness about responsible sexual practices in GBMSM. During the mpox outbreak, concerted efforts were made by public health agencies (eg, the UK Health Security Agency) to disseminate accurate scientific information about mpox to GBMSM in collaboration with community-based organisations and charities, the dating app Grindr, and organisers of Pride events,[30] building on existing communication channels.[31]

Therefore, knowledge and beliefs about mpox may be different in GBMSM and the general population. For example, it is likely that perceived worry and risk are higher in GBMSM—the population most affected by the mpox outbreak—than the general population.

There are few quantitative studies investigating mpox knowledge, attitudes and beliefs in the UK during the 2022 outbreak. Studies investigating the effectiveness of messages that could be used in official communications are particularly important. Sociodemographic characteristics associated with intended uptake of protective behaviours can help to identify groups that may benefit from increased messaging and support to adhere to public health guidelines. Psychological factors, such as knowledge, attitudes and beliefs, that are associated with intention to enact protective behaviours can provide insights into potentially modifiable factors that could be included in official messaging. Collecting this information forms the start of an evidence base for policy decisions. In this study, we recruited a general population sample and three GBMSM samples from: a market research company, Grindr and Meta (Facebook and Instagram). We investigated knowledge, attitudes and beliefs about mpox, and intentions for key behaviours that could prevent the spread of mpox (self-isolation, help seeking (as the advised route into testing), sexual contact behaviour when symptomatic, contact sharing, vaccination). We used an experimental approach to investigate the impact of different brief communication approaches (promoting perceived susceptibility to and severity of illness/necessity and efficacy of the response/benefits of the response/low perceived costs of response) on intended behaviour. Psychological and sociodemographic factors associated with intended uptake of behaviours were also investigated.

## METHOD
### Design
Online cross-sectional survey conducted by Savanta, a Market Research Society Company Partner. Data were collected between 5 September 2022 and 6 October 2022.

### Participants
Eligibility criteria for the general population sample were living in the UK and being aged 18 years or over. For the GBMSM samples, additional criteria were being a man and identifying as being gay, bisexual or having sex with men.

Recruitment for the general population sample used quota sampling, a standard opinion polling method that allows for rapid data collection. Members of Savanta's specialist research panel (n=150 000 across proprietary panels; people who have signed up to complete online surveys) were sent the survey link. Quota sampling uses predetermined targets, based on preselected sociodemographic characteristics (quotas) that match the national population. Participants who belong to a quota that has already been filled are prevented from completing

the survey. Therefore, response rate is not an accurate measure of response bias in quota samples. For this study, quotas were based on age, gender, socioeconomic grade and Government Office Region, and reflected targets based on 2020 mid-year estimates.[32]

We recruited three GBMSM samples (Savanta, Grindr, Meta (Facebook and Instagram)). A 'boost' sample of 250 GBMSM was recruited by Savanta, using the same quota sampling (excluding gender as all participants were male). We also recruited for the GBMSM sample through a push-notification inbox advert on Grindr (1.1 million adverts delivered, 24 933 opened and 4288 clicks) and using targeted adverts on Meta (308 472 adverts shown, 108 865 adverts seen and 3675 clicks). No quotas were placed on these samples.

### Study materials
Full study materials are in online supplemental materials 1. Items were based on previously validated measures,[33 34] and items used in previous surveys during the COVID-19 pandemic by our group.[11 14 35 36]

### Outcome measures
Self-isolation intention was measured using two items, asking participants to imagine that they were contacted by public health officials and told that they needed to self-isolate for 21 days because they had mpox, and because they had come into high-risk contact with a case. Responses were given on a five-point scale from 'definitely would not' to 'definitely would'. The order of items was randomised between participants to mitigate potential order effects. As answers on both items were highly correlated across all respondents (r=0.77, p=0.001), we combined responses to give a 9-point scale (2–10; summing responses 'definitely would not'=1 to 'definitely would'=5; higher number indicates greater intention).

To measure help-seeking intention, participants were asked to imagine that they developed an unexplained rash with blisters and learnt that they had come into contact with a mpox case. They were then asked what they would do, from a list of 10 behaviours including waiting to see if they got better, contacting healthcare services, letting people you had been in recent close contact know and searching for information. Responses for each item were given on the same 5-point 'definitely would not' to 'definitely would' scale. We created a binary variable, coding participants as intending to seek help immediately if they answered 'probably' or 'definitely would' to any help seeking where they would encounter a health professional (trying to book an appointment with a general practitioner, visiting a pharmacist/chemist, going to Accident & Emergency or another National Health Service (NHS) service, calling NHS111 or 999, visiting a walk-in sexual health clinic, or calling a sexual health clinic), and did not select 'wait a day or two to see if they get better or clear up on their own' versus did not select any help-seeking behaviour where they would encounter a health

professional, or selected a help-seeking behaviour but also stated that they would 'wait a day or two to see…'.

Participants were then asked about their intended contact behaviour in the same scenario, being asked 'in the following 21 days, realistically how much' they would come into skin-to skin contact with others, have sexual contact, have sex without using a condom, go to a crowded place, help or provide care for a vulnerable person, and go to a public place where they may come into physical contact with someone else. For each item, participants responded 'I would completely stop doing this', 'less than normal', 'same as normal', 'more than normal', 'not applicable, I wouldn't do this anyway', 'don't know' or 'prefer not to say'. We focused our analyses on the item asking about sexual contact (from kissing to intercourse) with other people, recoding it into a binary item ('I would completely stop doing this' vs would do this less than, same as, or more than usual). Answers of 'not applicable, I wouldn't do this anyway', 'don't know' and 'prefer not to say' were coded as missing.

We measured intention to share details of close contacts by asking participants to indicate how likely they were, if asked by public health officials, to share contact details of every person who had been in their home, they had had sexual contact with, they had skin-to-skin contact with, and that they had shared bedding, towels or clothes with in the last 7 days, and every place they had had sexual contact with someone. Responses were given on a 5-point 'definitely would not' (1) to 'definitely would' (5) scale. We used the most recent official guidance on contact tracing available at the start of data analysis to select the item most relevant to contact tracing efforts (identifying every person you had sexual contact with in the last 7 days).[7]

Participants were asked if they had received a smallpox vaccine in 2022. Those who indicated they had not had a vaccine were asked about their vaccination intention. We asked participants how likely they would be to have a smallpox vaccine if they were offered one. Responses were given on a 5-point 'definitely would not' (1) to 'definitely would' (5) scale.

## Motivational messaging

Very brief motivational messages were constructed based on components of the PMT.[13] The general population samples were randomised to one of four groups and were shown messages about: (1) perceived risk of illness plus necessity and efficacy of the response, (2) perceived risk of illness plus benefits of the response, (3) perceived risk of illness plus low perceived costs of response and (4) a control message of similar length giving factual details about the mpox outbreak. Messages are shown in online supplemental materials 1. Messages to promote perceived risk of illness (ie, susceptibility to and severity of mpox) were included in all motivational messages, as we hypothesised that perceiving a risk is necessary before choosing to adopt a behaviour to mitigate that risk. Due to anticipated smaller sample sizes, the GBMSM samples were randomised to one of two messages. The first included all motivational components, whereas the second, a control message, gave factual information about the mpox outbreak.

## Psychological factors

We asked participants how much they had seen or heard about mpox, how worried they were about mpox, and how much risk they thought mpox posed to people in the UK and themselves personally. For these items, answers of 'don't know' were coded as missing.

Participants were also asked about their perceived susceptibility to mpox (two items: would be likely to come into contact with a case; would be likely to catch mpox if in contact with a case) and perceived severity of mpox. We asked participants how much they agreed that: their personal behaviour had an impact on the spread of mpox; their life had been negatively affected by changes made in response to the mpox outbreak; the risks of mpox were being exaggerated; people who catch mpox usually make a full recovery without treatment; and that mpox is only a risk to men who are gay, bisexual or have sex with men. For perceptions such as these, with no right or wrong answers, we recoded answers of 'don't know' as the midpoint on the scale.

To measure perceived knowledge, we asked participants three items about whether they had a good idea how people catch mpox, they knew the main symptoms of mpox, and they thought it would be easy to tell if someone had mpox. Knowledge about mpox symptoms was measured using a question asking participants to identify the main symptoms of mpox from a list of 15 taken from the NHS mpox website[1] and common, non-specific symptoms. Participants could select up to four symptoms. Understanding of transmission was measured using seven items, asking about contact and droplet transmission (adapted from Rubin et al,[33] and other modes of transmission as specified by the WHO website[37]). For factual questions such as these, we coded 'don't know' as incorrect.

Behaviour-specific perceptions were also investigated. We used a series of 10 items to measure factors that may be associated with self-isolation, including perceived effectiveness, social norms, having the necessary support and impact on social connectedness, family well-being and finances. Factors that may affect intention to seek help were measured by six items asking participants to what extent they agreed that they would not want to know the results of a mpox test, they would be worried what their friends, family or employer thought of them if they had mpox, not wanting to have a mpox test on their medical record, testing is an effective way to prevent the spread of mpox, and being willing to contact a sexual health clinic if they thought they had mpox symptoms or had come into contact with a case. Vaccination attitudes were measured by eight items, asking about general vaccine attitudes, perceived social norms, perceived effectiveness of vaccination, worry about vaccine side effects, that the vaccine

could make you infectious to others, and thinking that those who come into high-risk contact with mpox should be vaccinated. Responses of 'don't know' were recoded to the midpoint of the scale.

### Sociodemographic characteristics

Participants were asked to report their age, gender,[38] sexual orientation,[39] socioeconomic grade,[40] financial hardship (adapted from Organisation for Economic Co-operation and Development[41]), employment status, highest level of education, ethnicity, marital status, how many people lived in their household, whether they were the parent or guardian of any dependent children, and if they had any pets. Questions asking about gender and the categorisations used were based on those used by the Genitourinary Medicine Clinic Activity Dataset sexually transmitted infections (STI) surveillance system in England.[38] Those who were employed were asked if they were a frontline health or social care worker and if they needed to leave home for work. For these items, we coded participants who were not employed as not being a frontline health or social care worker (prefer not to say coded as missing) and not needing to leave home for work, respectively. Participants were asked for their full postcode, from which region and indices of multiple deprivation were determined.[42]

We asked participants if they or a household member had a chronic illness, whether they were pregnant, had ever taken pre-exposure prophylaxis (PrEP) for HIV, and for their vaccination status for smallpox (in 2022 and before 2022), hepatitis A and COVID-19 (two doses or more).

Participants were also asked how many male and female sexual partners they had had in the last 3 weeks and last 3 months.

### Patient and public involvement

To ensure the research aims and study information were appropriate, members of the public were involved in the development of the funding application and survey materials. For the funding application, six people gave feedback on the initial proposal resulting in changes to aims of the study and terminology used. For the survey materials, four lay people (two GBMSM) gave feedback on the questionnaire and motivational messages, resulting in changes to survey items and messages to improve clarity, validity and readability of statements.

### Power

A sample size of 3000 allows a 95% CI of plus or minus 1.8% for the prevalence estimate for a survey item with a prevalence of around 50% (sample size of 250 gives a 95% CI of plus or minus 6.2%; sample size of 1000 gives a 95% CI of plus or minus 3.1%).

For multiple linear regression analyses, a sample of 830 allowed over 99% power to detect small effect sizes (f=0.10) at p=0.001 (43 predictors). For logistic regression analyses, a sample of 830 allowed over 99% power to detect small effect sizes (OR=1.68[43]) at p=0.001 (42.5% self-isolation, 18.0% requesting a test, 79.1% sharing details of close contacts).[11]

### Analysis

Information about data preparation is reported in online supplemental materials 2.

We tested whether sociodemographic characteristics of GBMSM samples were different depending on the recruitment method (Savanta, Grindr, Meta). Due to significant differences, further analyses were conducted in each sample separately.

First, we tested the influence of motivational messages on outcomes using $\chi^2$ tests (binary outcomes), one-way analyses of variance (general population sample, continuous outcomes) or t-tests (GBMSM, continuous outcomes). For vaccination, we investigated the influence of motivational messages on intention to be vaccinated if advised (excluding people already vaccinated).

Next, we investigated psychological and contextual factors associated with intended behaviours (self-isolation, help seeking, sexual contact, sharing details of contacts, vaccination). To minimise analyses conducted, we investigated one variable per outcome, except for vaccination (GBMSM sample investigated two outcomes). For vaccination, as smallpox vaccine uptake in GBMSM was high, we used two binary outcomes: vaccination uptake in 2022 (vaccinated vs not vaccinated), and a computed variable indicating vaccine intention and uptake (vaccinated or intend to be vaccinated ('definitely' or 'probably would') vs not vaccinated and do not intend to be vaccinated ('not sure', 'probably would not', 'definitely would not')). For the general population sample, we used only the computed variable. We conducted regressions (binary logistic for binary outcomes, linear for continuous outcomes) in the general population sample and Grindr sample (the target GBMSM sample who were actively seeking new partners; Savanta sample excluded due to small numbers; Meta sample excluded as they differed significantly from the general population).

We entered variables into regressions in blocks. In the first block, we entered sociodemographic characteristics: gender (general population sample only, male/female), sexual orientation (general population sample only, straight or heterosexual/gay, lesbian, bisexual or queer), age, region, having a dependent child in household (no/ yes), employment status (working/not working), education (GCSE (General Certificate of Secondary Education), vocational, A-level, no formal qualifications/degree or higher), ethnicity (white British/white other/black, Asian, other minoritised ethnicity), marital status, living alone, having a chronic illness oneself (none/present), Index of Multiple Deprivation (deciles), socioeconomic grade (ABC1/C2DE), financial hardship and motivational messages. In the Grindr sample, we also included smallpox vaccination status in 2022 (except for vaccine uptake outcome) and ever having taken PrEP for HIV.

In the second block, we entered psychological and contextual factors: self-reported knowledge, knowledge of mpox symptoms, knowledge of mpox transmission, amount heard about mpox, worry about mpox, perceived risk of mpox (to oneself and to people in the UK), perceived susceptibility and severity of mpox, thinking that you are immune to mpox, that your personal behaviour has an impact on how mpox spreads, that your life has been negatively affected by changes made in response to the mpox outbreak, that the risks of mpox are being exaggerated, that mpox is only a risk to men who are gay, bisexual or have sex with men, and that people who catch mpox usually make a full recovery even without treatment.

For self-isolation, help seeking and vaccination, a third block was also added, which included specific factors potentially associated with individual outcomes. Items were chosen through principal component analyses (see online supplemental materials 3).

All analyses were carried out in SPSS V.28. Data are unweighted.

Many comparisons were investigated in regression models (n=40 to n=43, based on outcome). Therefore, we applied a conservative Bonferroni correction and only reported as significant results with p≤0.001 to reduce the risk of type I errors.

## RESULTS

Top-line results for all survey materials, by sample, are shown in online supplemental materials 1. Anonymised data are available online.[44]

For regression analyses, we report imputed values. Results using imputed values were compared with non-imputed data. There were no substantial differences in results with and without imputed values.

### Participant characteristics

There were significant differences in participant characteristics by sampling method. Most notably, participants recruited from Grindr and Meta were more likely to be working, need to leave home for work, more highly educated, higher socioeconomic grade and have less financial hardship (table 1). In the general population sample, participants were mostly female (57%), white British (87%), with a mean age of 49 years.

### Motivational messaging

There was no effect of motivational messaging on outcomes, except for in the sample recruited from Meta (see online supplemental materials 4). In this sample, those receiving the motivational messages were less likely to intend to self-isolate for 21 days ($t(1034)=-2.81$, p=0.005; motivational message, n=529, M=7.1, SD=2.3; control message, n=507, M=7.5, SD=2.2), share details of all recent sexual contacts ($t(1029)=-2.05$, p=0.04; motivational message, n=526, M=4.1, SD=1.2; control message, n=505, M=4.2, SD=1.1) and be vaccinated for smallpox

if advised ($t(612)=-2.21$, p=0.03; motivational message, n=304, M=4.7, SD=0.9; control message, n=310, M=4.8, SD=0.6).

### Self-isolation

Rates of intended self-isolation were higher when imagining you were a case than a high-risk contact. Three-quarters of the general population sample intended to self-isolate for 21 days if they were to develop mpox (75.2%, 95% CI 73.7% to 76.7%, n=2294; table 2). However, only 68.9% (95% CI 67.2% to 70.5%, n=2100) intended to self-isolate if they were to come into contact with a case. Intention to self-isolate was lower in Grindr and Meta samples.

In the general population, intention to self-isolate was associated with: less financial hardship, being more worried about mpox, perceiving a bigger threat of mpox to people in the UK, greater perceived susceptibility and severity of mpox, and greater perceived social norms for self-isolation (tables 3 and 4). Not intending to self-isolate was associated with agreeing that if you had mpox symptoms, you would not want to tell anyone as you did not want to self-isolate, believing the risks of mpox were being exaggerated, and that if you had to self-isolate due to mpox it would have a negative impact on your work (table 4). In the Grindr sample, self-isolation intention was associated with greater perceived social norms (table 4). Not intending to self-isolate was associated with agreeing that if you had mpox symptoms, you would not want to tell anyone as you did not want to self-isolate, and that if you had to self-isolate due to mpox it would have a negative impact on your work (table 4).

### Help seeking

Approximately half of participants in the general population, Savanta GBMSM and Grindr samples indicated that they would seek help immediately (95% CI 49.2% to 53.3%; table 2). Intention to seek help immediately was higher in the Meta sample (62.3%, 95% CI 59.3 to 65.2, n=645).

In the general population, intention to seek help immediately was associated with being older (aOR (adjusted Odds Ratio) 1.012, 95% CI 1.006 to 1.019, p<0.001), disagreeing that the risks of mpox are being exaggerated (aOR 0.83, 95% CI 0.76 to 0.91, p<0.001), and being willing to contact a sexual health clinic if you thought you had mpox symptoms or had been in contact with a case (aOR 1.25, 95% CI 1.16 to 1.34, p<0.001; online supplemental materials 5). In the Grindr sample, intention to seek help immediately was associated with being willing to contact a sexual health clinic if you thought you had mpox symptoms or had been in contact with a case (aOR 1.57, 95% CI 1.29 to 1.91, p<0.001; online supplemental materials 5).

### Sexual contact behaviour when symptomatic

In the general population and Savanta GBMSM, 77.2% (95% CI 75.6% to 78.9%, n=1923) and 79.2% (73.7%

**Table 1** Participant characteristics, by recruitment method

| | | General population, n=3050 | GBMSM samples | | | |
| | | | Savanta GBMSM, n=247 | Grindr, n=831 | Meta, n=1036 | P value |
|---|---|---|---|---|---|---|
| Gender | Male (including transman) | 1278 (41.9) | 238 (96.4) | 828 (99.6) | 1013 (97.8) | – |
| | Gender same as assigned at birth (no or prefer not to say) | 11 (0.4) | 9 (3.6) | 3 (0.04) | 23 (2.2) | – |
| | Female (including transwoman) | 1729 (56.7) | – | – | – | – |
| | Gender same as assigned at birth (no or prefer not to say) | 7 (0.02) | – | – | – | – |
| | Non-binary | 8 (0.3) | – | – | – | – |
| | Gender same as assigned at birth (no or prefer not to say) | 9 (0.3) | – | – | – | – |
| Sexual orientation | Straight or heterosexual | 2795 (92.7) | – | – | – | – |
| | Gay, lesbian, bisexual or queer | 221 (7.3) | 247 (100.0) | 831 (100.0) | 1036 (100.0) | – |
| Age | Range 18–98 years | M=48.6, SD=17.4 | M=47.1, SD=16.5 | M=44.2, SD=12.5 | M=47.6, SD=11.9 | <0.001* |
| Region† | Midlands (East and West) | 522 (17.1) | 39 (15.8) | 73 (8.8) | 85 (8.2) | <0.001* |
| | South (East, West, East of England) | 992 (32.5) | 70 (28.3) | 207 (24.9) | 279 (26.9) | |
| | North (East, West, Yorkshire and the Humber) | 774 (25.4) | 58 (23.5) | 119 (14.3) | 155 (15.0) | |
| | London | 308 (10.1) | 46 (18.6) | 195 (23.5) | 379 (36.6) | |
| | Devolved nations (Scotland, Wales and Northern Ireland) | 454 (14.9) | 34 (13.8) | 95 (11.4) | 84 (8.1) | |
| | Not specified | 0 (0.0) | 0 (0.0) | 142 (17.1) | 54 (5.2) | |
| Dependent child in household | No | 2075 (68.0) | 210 (85.0) | 796 (95.8) | 1014 (97.9) | <0.001* |
| | Yes | 975 (32.0) | 37 (15.0) | 35 (4.2) | 22 (2.1) | |
| Employment status | Not working | 1286 (42.6) | 93 (38.3) | 157 (19.0) | 201 (19.5) | <0.001* |
| | Working | 1736 (57.4) | 150 (61.7) | 669 (81.0) | 831 (80.5) | |
| Front-line health or social care worker | No | 2668 (88.3) | 224 (90.7) | 734 (88.6) | 909 (87.9) | 0.66 |
| | Yes | 355 (11.7) | 23 (9.3) | 94 (11.4) | 125 (12.1) | |
| Need to leave home for work | Do not need to leave home for work | 1946 (63.8) | 149 (60.3) | 371 (44.6) | 499 (48.2) | <0.001* |
| | Need to leave home for work | 1104 (36.2) | 98 (39.7) | 460 (55.4) | 537 (51.8) | |

Continued

**Table 1** Continued

| | | General population, n=3050 | GBMSM samples | | | |
| | | | Savanta GBMSM, n=247 | Grindr, n=831 | Meta, n=1036 | P value |
|---|---|---|---|---|---|---|
| Education | GCSE/vocational/ A-level/No formal qualifications | 2065 (67.7) | 159 (64.4) | 295 (35.5) | 253 (24.4) | <0.001* |
| | Degree or higher (Bachelors, Masters, PhD) | 985 (32.3) | 88 (35.6) | 536 (64.5) | 783 (75.6) | |
| Ethnicity | White British | 2649 (87.1) | 214 (86.6) | 629 (76.2) | 799 (78.0) | <0.001* |
| | White other | 116 (3.8) | 18 (7.3) | 117 (14.2) | 163 (15.9) | |
| | Black, Asian, other minoritised ethnicity | 275 (9.0) | 15 (6.1) | 80 (9.7) | 63 (6.1) | |
| Marital status | Not partnered | 1265 (41.6) | 141 (57.3) | 561 (68.3) | 486 (47.3) | <0.001* |
| | Partnered | 1775 (58.4) | 105 (42.7) | 260 (31.7) | 541 (52.7) | |
| Live alone | Live with someone else | 2387 (78.3) | 146 (59.1) | 473 (56.9) | 656 (63.3) | 0.02 |
| | Live alone | 663 (21.7) | 101 (40.9) | 358 (43.1) | 380 (36.7) | |
| Own chronic illness | None | 2207 (72.4) | 167 (67.6) | 619 (74.5) | 731 (70.6) | 0.05 |
| | Present | 843 (27.6) | 80 (32.4) | 212 (25.5) | 305 (29.4) | |
| Household member chronic illness | None | 2659 (87.2) | 226 (91.5) | 743 (89.4) | 921 (88.9) | 0.49 |
| | Present | 391 (12.8) | 21 (8.5) | 88 (10.6) | 115 (11.1) | |
| Ever taken PrEP for HIV | No | 2993 (98.6) | 216 (88.5) | 427 (51.5) | 547 (52.9) | <0.001* |
| | Yes | 42 (1.4) | 28 (11.5) | 402 (48.5) | 487 (47.1) | |
| Vaccinated for smallpox in 2022 | Not vaccinated | 2888 (94.7) | 221 (89.5) | 566 (68.1) | 614 (59.3) | <0.001* |
| | Vaccinated | 162 (5.3) | 26 (10.5) | 265 (31.9) | 422 (40.7) | |
| Index of Multiple Deprivation† | Deciles (1st=most deprived, 10th=least deprived) | M=4.5, SD=2.9 | M=4.4, SD=3.0 | N=689, M=4.7, SD=2.8 | N=982, M=4.9, SD=2.7 | <0.001* |
| Socioeconomic grade | ABC1 | 1796 (58.9) | 169 (68.4) | 694 (83.5) | 944 (91.1) | <0.001* |
| | C2DE | 1254 (41.1) | 78 (31.6) | 137 (16.5) | 92 (8.9) | |
| Financial hardship† | 4 (lowest hardship) to 13 (most hardship) | N=2743, M=6.1, SD=2.3 | N=223, M=5.9, SD=2.4 | N=784, M=5.2, SD=1.8 | N=1004, M=4.7, SD=1.3 | <0.001* |

*p≤0.001.
†Using original (not imputed) data.
GBMSM, gay, bisexual or other men who have sex with men; GCSE, General Certificate of Secondary Education; PrEP, pre-exposure prophylaxis.

to 84.6%, n=171) intended to completely stop sexual contact if they were to develop an unexplained rash with blisters and learn that they had come into contact with a mpox case (table 2). Rates of intending to completely stop sexual contact were significantly higher in Grindr (91.3%, 95% CI 89.3% to 93.2%, n=721) and Meta samples (93.0%, 95% CI 91.4% to 94.6%, n=922). The number of days that participants would wait before resuming sexual contact from the start of their symptoms was also higher in Grindr and Meta samples than in general population and Savanta GBMSM samples ($F(3,3900)=29.0$, p<0.001;

general population: n=2021, M=15.7, SD=15.3; Savanta GBMSM: n=190, M=16.2, SD=14.4; Grindr: n=746, M=20.4, SD=15.8; Meta: n=947, M=20.0, SD=12.3).

In the general population, intending to completely stop any sexual contact if symptomatic was associated with being female, older, less financial hardship and being more knowledgeable about mpox transmission (tables 5 and 6). Not intending to completely stop sexual behaviour was associated with preferring not to say how many recent sexual partners you had had, thinking that you were already immune to mpox, that your life had

**Table 2** Main behavioural outcomes, by sample

| | | N (%, 95% CI) | | | | P value |
|---|---|---|---|---|---|---|
| | | General population, total n=3050 | Savanta GBMSM, total n=247 | Grindr, total n=831 | Meta, total n=1036 | |
| Self-isolation for 21 days | If 'you have' mpox — Probably or definitely would not, or not sure | 756 (24.8, 23.3 to 26.3) | 53 (21.5, 16.3 to 26.6) | 246 (29.6, 26.5 to 32.7) | 275 (26.5, 23.9 to 29.2) | – |
| | Probably or definitely would | 2294 (75.2, 73.7 to 76.7) | 194 (78.5, 73.4 to 83.7) | 585 (70.4, 67.3 to 73.5) | 761 (73.5, 70.8 to 76.1) | |
| | If 'you have come into high-risk contact with someone who has' mpox — Probably or definitely would not, or not sure | 950 (31.1, 29.5 to 32.8) | 77 (31.2, 25.4 to 37.0) | 371 (44.6, 41.3 to 48.0) | 491 (47.4, 44.3 to 50.4) | – |
| | Probably or definitely would | 2100 (68.9, 67.2 to 70.5) | 170 (68.8, 63.0 to 74.6) | 460 (55.4, 52 to 58.7) | 545 (52.6, 49.6 to 55.7) | |
| | Sum of intention if case and high-risk contact — 2 (lowest intention) to 10 (highest intention) | M=8.0, SD=2.2 | M=7.9, SD=2.3 | M=7.2, SD=2.4 | M=7.3, SD=2.3 | <0.001* |
| Help seeking | Any action that involves contacting a healthcare professional (by phone or in person) — Would not seek help or would 'wait and see' for a day or two to see if symptoms resolved | 1548 (50.8, 49.0 to 52.5) | 125 (50.6, 44.3 to 56.9) | 388 (46.7, 43.3 to 50.1) | 391 (37.7, 34.8 to 40.7) | <0.001* |
| | Would seek help immediately | 1502 (49.2, 47.5 to 51.0) | 122 (49.4, 43.1 to 55.7) | 443 (53.3, 49.9 to 56.7) | 645 (62.3, 59.3 to 65.2) | |
| Sexual contact behaviour† | Have sexual contact with others — Would not completely stop | 567 (22.8, 21.1 to 24.4) | 45 (20.8, 15.4 to 26.3) | 69 (8.7, 6.8 to 10.7) | 69 (7.0, 5.4 to 8.6) | <0.001* |
| | Completely stop | 1923 (77.2, 75.6 to 78.9) | 171 (79.2, 73.7 to 84.6) | 721 (91.3, 89.3 to 93.2) | 922 (93.0, 91.4 to 94.6) | |
| Contact sharing | Sexual contacts — Probably or definitely would not, or not sure | 697 (23.3, 21.7 to 24.8) | 60 (24.7, 19.2 to 30.2) | 167 (20.3, 17.5 to 23.0) | 204 (19.8, 17.4 to 22.2) | – |
| | Probably or definitely would | 2300 (76.7, 75.2 to 78.3) | 183 (75.3, 69.8 to 80.8) | 656 (79.7, 77 to 82.5) | 827 (80.2, 77.8 to 82.6) | |
| | Scale — 1 (definitely would not) to 5 (definitely would) | N=2997, M=4.2, SD=1.2 | N=243, M=4.1, SD=1.2 | N=823, M=4.1, SD=1.2 | N=1031, M=4.2, SD=1.1 | 0.57 |
| Vaccination intention | Vaccinated in 2022 — No or not sure | 2888 (94.7, 93.9 to 95.5) | 221 (89.5, 85.6 to 93.3) | 566 (68.1, 64.9 to 71.3) | 614 (59.3, 56.3 to 62.3) | <0.001* |
| | Yes | 162 (5.3, 4.5 to 6.1) | 26 (10.5, 6.7 to 14.4) | 265 (31.9, 28.7 to 35.1) | 422 (40.7, 37.7 to 43.7) | |
| | Vaccinated in 2022 or would be vaccinated if advised — Not vaccinated and probably or definitely would not be vaccinated or not sure | 786 (25.8, 24.2 to 27.3) | 37 (15.0, 10.5 to 19.5) | 54 (6.5, 4.8 to 8.2) | 38 (3.7, 2.5 to 4.8) | <0.001* |
| | Vaccinated or probably or definitely would be vaccinated | 2264 (74.2, 72.7 to 75.8) | 210 (85.0, 80.5 to 89.5) | 777 (93.5, 91.8 to 95.2) | 998 (96.3, 95.2 to 97.5) | |

*p≤0.001.
†Answers of 'don't know', 'prefer not to say' and 'not applicable, I wouldn't do this anyway' were coded as missing, therefore total ns are substantially lower (general population, n=2490; Savanta GBMSM, n=216; Grindr, n=790; Meta, n=991).
GBMSM, gay, bisexual or other men who have sex with men.

**Table 3** Associations between intending to self-isolate and sociodemographic characteristics and motivational message, by sample

| Participant characteristics | Level | General population | | Grindr | |
|---|---|---|---|---|---|
| | | B (95% CI) | P value | B (95% CI) | P value |
| Gender | Male (including transman) | Ref | – | – | – |
| | Female (including transwoman) | 0.15 (0.01 to 0.30) | 0.03 | – | – |
| Sexual orientation | Straight or heterosexual | Ref | – | – | – |
| | Gay, lesbian, bisexual or queer | 0.06 (−0.21 to 0.33) | 0.67 | – | – |
| Age | Range 18–98 years | 0.01 (0.00 to 0.01) | 0.06 | 0.00 (−0.01 to 0.02) | 0.66 |
| | Quadratic term (age–mean)$^2$ | 0.0002 (−0.0001 to 0.0005) | 0.17 | 0.001 (0.000 to 0.002) | 0.02 |
| Region | Midlands (East and West) | Ref | – | Ref | – |
| | South (East, West, East of England) | 0.19 (−0.39 to 0.01) | 0.07 | 0.02 (−0.51 to 0.55) | 0.95 |
| | North (East, West, Yorkshire and the Humber) | 0.07 (−0.28 to 0.13) | 0.49 | 0.06 (−0.53 to 0.64) | 0.85 |
| | London | 0.11 (−0.39 to 0.16) | 0.42 | 0.01 (−0.59 to 0.57) | 0.98 |
| | Devolved nations (Scotland, Wales and Northern Ireland) | 0.04 (−0.30 to 0.22) | 0.78 | 0.48 (−0.20 to 1.16) | 0.17 |
| Dependent child in household | No | Ref | – | Ref | – |
| | Yes | 0.15 (−0.32 to 0.03) | 0.10 | 0.42 (−1.13 to 0.29) | 0.25 |
| Employment status | Not working | Ref | – | Ref | – |
| | Working | 0.09 (−0.12 to 0.31) | 0.39 | 0.13 (−0.61 to 0.34) | 0.58 |
| Frontline health or social care worker | No | Ref | – | Ref | – |
| | Yes | 0.08 (−0.14 to 0.30) | 0.47 | 0.68 (0.24 to 1.13) | 0.003 |
| Need to leave home for work | Do not need to leave home for work | Ref | – | Ref | – |
| | Need to leave home for work | 0.26 (−0.45 to −0.07) | 0.007 | 0.14 (−0.22 to 0.51) | 0.44 |
| Education | GCSE/vocational/A-level/no formal qualifications | Ref | – | Ref | – |
| | Degree or higher (Bachelors, Masters, PhD) | 0.02 (−0.13 to 0.18) | 0.77 | 0.16 (−0.48 to 0.16) | 0.31 |
| Ethnicity | White British | Ref | – | Ref | – |
| | White other | 0.23 (−0.59 to 0.13) | 0.21 | 0.33 (−0.75 to 0.09) | 0.12 |
| | Black, Asian, other minoritised ethnicity | 0.11 (−0.15 to 0.37) | 0.40 | 0.03 (−0.52 to 0.46) | 0.91 |
| Marital status | Not partnered | Ref | – | Ref | – |
| | Partnered | 0.15 (−0.04 to 0.34) | 0.12 | 0.13 (−0.24 to 0.49) | 0.50 |
| Live alone | Live with someone else | Ref | – | Ref | – |
| | Live alone | 0.06 (−0.15 to 0.28) | 0.56 | 0.07 (−0.40 to 0.26) | 0.69 |
| Own chronic illness | None | Ref | – | Ref | – |
| | Present | 0.21 (0.05 to 0.38) | 0.009 | 0.10 (−0.24 to 0.44) | 0.56 |
| Ever taken PrEP for HIV | No | – | – | Ref | – |
| | Yes | – | – | 0.11 (−0.43 to 0.22) | 0.51 |
| Vaccinated for smallpox in 2022 | Not vaccinated | – | – | Ref | – |
| | Vaccinated | – | – | 0.03 (−0.32 to 0.39) | 0.86 |
| Index of Multiple Deprivation | Deciles (1st=most deprived, 10th=least deprived) | 0.01 (−0.04 to 0.02) | 0.48 | 0.02 (−0.05 to 0.08) | 0.66 |
| Socioeconomic grade | ABC1 | Ref | – | Ref | – |
| | C2DE | 0.09 (−0.24 to 0.06) | 0.24 | 0.03 (−0.46 to 0.39) | 0.88 |

Continued

**Table 3** Continued

| Participant characteristics | Level | General population | | Grindr | |
|---|---|---|---|---|---|
| | | B (95% CI) | P value | B (95% CI) | P value |
| Financial hardship | 4 (lowest hardship) to 13 (most hardship) | 0.08 (−0.12 to −0.05) | <0.001* | 0.02 (−0.11 to 0.07) | 0.70 |
| Total no of sexual partners (male and female) in last 3 weeks | 0 | Ref | – | – | – |
| | 1 | 0.04 (−0.23 to 0.14) | 0.65 | – | – |
| | 2–4 | 0.25 (−0.59 to 0.10) | 0.16 | – | – |
| | 5 or more | 0.09 (−0.69 to 0.52) | 0.77 | – | – |
| | Prefer not to say | 0.22 (−0.42 to −0.02) | 0.03 | – | – |
| No of male sexual partners in last 3 weeks | 0 | – | – | Ref | – |
| | 1 | – | – | 0.50 (−0.92 to −0.08) | 0.02 |
| | 2–4 | – | – | 0.09 (−0.47 to 0.30) | 0.65 |
| | 5–9 | – | – | 0.48 (−1.00 to 0.03) | 0.07 |
| | 10 or more | – | – | 0.32 (−0.96 to 0.33) | 0.34 |
| | Prefer not to say | – | – | 0.83 (0.12 to 1.53) | 0.02 |
| Motivational message | Perceived risk of illness and necessity and efficacy of the response | 0.06 (−0.25 to 0.12) | 0.50 | – | – |
| | Perceived risk of illness and benefits of the response | 0.09 (−0.28 to 0.09) | 0.33 | – | – |
| | Perceived risk of illness and low perceived costs of response | 0.19 (−0.38 to 0.00) | 0.05 | – | – |
| | Control | Ref | – | – | – |
| Motivational message | All motivational components | – | – | 0.10 (−0.37 to 0.18) | 0.48 |
| | Control | – | – | Ref | – |

A higher score indicates greater intention to self-isolate. Variables were entered into the linear regression model in blocks (block 1: sociodemographic variables and motivational message, block 2: psychological factors, block 3: isolation-specific beliefs). Results for block 3, using pooled estimates are reported.
*p≤0.001.
GCSE, General Certificate of Secondary Education; PrEP, pre-exposure prophylaxis.

been negatively affected by changes made in response to the mpox outbreak, and that the risks of mpox were being exaggerated (tables 5 and 6). No associations reached significance (Bonferroni corrected) in the Grindr sample.

### Sharing details of contacts

There was no difference in intention to share details of all sexual partners in the last 7 days between samples (table 2), with 75.3%–80.2% saying that they probably or definitely would.

In the general population sample, intention to share details of every sexual contact in the last 7 days was associated with being female, older, less financial hardship, higher perceived susceptibility to and severity of mpox, and higher knowledge about modes of mpox transmission (tables 7 and 8). Not intending to share details of every recent sexual contact was associated with preferring not to say how many recent sexual partners you had had, thinking that your life had been negatively affected by changes made in response to the mpox outbreak, and thinking that the risks of mpox had been exaggerated

(tables 7 and 8). No associations reached our threshold for significance in the Grindr sample.

### Vaccination

Few people had been vaccinated for smallpox in 2022 in the general population sample (5.3%, 95% CI 4.5% to 6.1%, n=162; table 2). This was significantly higher in GBMSM samples (10.5% to 40.7%). In a measure of actual and intended vaccination, 96.3% (95% CI 95.2% to 97.5%, n=998) of the Meta sample and 93.5% (95% CI 91.8% to 95.2%, n=777) of the Grindr sample were vaccinated or intended to be vaccinated for smallpox if advised. Rates were significantly lower in the general population and Savanta GBMSM samples.

In the general population, being vaccinated for smallpox in 2022 or intending to be vaccinated if offered a vaccine was associated with being older (aOR 1.015, 95% CI 1.007 to 1.023, p<0.001), more worried about mpox (1.49, 95% CI 1.24 to 1.80, p<0.001), and perceiving a higher susceptibility to and severity of mpox (1.31, 95% CI 1.11 to 1.55, p<0.001; online supplemental materials 6). Not

**Table 4** Associations between intending to self-isolate and psychological factors and isolation-specific beliefs, by sample

| Factor | Level | General population | | Grindr | |
|--------|-------|--------------------|--|--------|--|
| | | B (95% CI) | P value | B (95% CI) | P value |
| Amount heard about mpox | I have not seen or heard anything (1) to I have seen or heard a lot (3) | 0.09 (−0.07 to 0.24) | 0.26 | 0.09 (−0.41 to 0.23) | 0.57 |
| Worry about mpox | Not at all worried (1) to extremely worried (4) | 0.26 (0.13 to 0.39) | <0.001* | 0.41 (0.12 to 0.71) | 0.006 |
| Perceived risk of mpox to oneself | No risk at all (1) to very high risk (5) | 0.05 (−0.17 to 0.06) | 0.39 | 0.25 (−0.46 to −0.04) | 0.02 |
| Perceived risk of mpox to people in UK | No risk at all (1) to very high risk (5) | 0.19 (0.07 to 0.31) | 0.001* | 0.030 (−0.20 to 0.25) | 0.82 |
| Perceived susceptibility and severity | Lowest (1) to highest (5) | 0.03 (0.19 to 0.42) | <0.001* | 0.12 (−0.13 to 0.37) | 0.36 |
| I am already immune to mpox | Strongly disagree, disagree, neither agree nor disagree, don't know | Ref | – | Ref | – |
| | Strongly agree and agree | 0.27 (0.03 to 0.52) | 0.03 | 0.12 (−0.61 to 0.36) | 0.62 |
| People who catch mpox usually make a full recovery, even if they do not receive any treatment | Strongly disagree (1) to strongly agree (5) | 0.05 (−0.14 to 0.03) | 0.23 | 0.26 (−0.43 to −0.08) | 0.004 |
| My personal behaviour has an impact on how mpox spreads | Strongly disagree (1) to strongly agree (5) | 0.09 (0.03 to 0.15) | 0.004 | 0.08 (−0.06 to 0.22) | 0.26 |
| My life has been negatively affected by changes made in response to the mpox outbreak | Strongly disagree (1) to strongly agree (5) | 0.01 (−0.09 to 0.07) | 0.89 | 0.03 (−0.17 to 0.11) | 0.70 |
| The risks of mpox are being exaggerated | Strongly disagree (1) to strongly agree (5) | 0.19 (−0.27 to −0.11) | <0.001* | 0.07 (−0.24 to 0.09) | 0.39 |
| Mpox is only a risk to men who are gay, bisexual or have sex with men | Strongly disagree (1) to strongly agree (5) | 0.02 (−0.08 to 0.05) | 0.58 | 0.17 (−0.32 to −0.02) | 0.03 |
| Perceived knowledge | Lowest (0) to highest (3) | 0.01 (−0.08 to 0.07) | 0.88 | 0.18 (−0.37 to 0.02) | 0.08 |
| Knowledge of mpox symptoms | Identified no symptoms (0) to identified four symptoms (4) | 0.08 (0.03 to 0.14) | 0.003 | 0.06 (−0.06 to 0.19) | 0.31 |
| Knowledge of mpox transmission | Lowest (0) to highest (6) | 0.02 (−0.03 to 0.07) | 0.42 | 0.04 (−0.10 to 0.17) | 0.59 |
| If I had mpox symptoms, I wouldn't want to tell anyone as I don't want to self-isolate | Strongly disagree (1) to strongly agree (5) | 0.66 (−0.73 to −0.60) | <0.001* | 1.01 (−1.15 to −0.87) | <0.001* |
| Most people would self-isolate if they were told to | Strongly disagree (1) to strongly agree (5) | 0.34 (0.28 to 0.41) | <0.001* | 0.41 (0.28 to 0.54) | <0.001* |
| If I had to self-isolate because I had tested positive for mpox…it would have a negative impact on my work | Strongly disagree (1) to strongly agree (5) | 0.15 (−0.21 to −0.09) | <0.001* | 0.34 (−0.46 to −0.21) | <0.001* |

A higher score indicates greater intention to self-isolate. Variables were entered into the linear regression model in blocks (block 1: sociodemographic variables and motivational message, block 2: psychological factors, block 3: isolation-specific beliefs). Results for block 3, using pooled estimates are reported.
*p≤0.001.

intending to be vaccinated was associated with needing to leave home for work (aOR 0.56, 95% CI 0.43 to 0.73, p<0.001) and agreeing that the risks of mpox were being exaggerated (aOR 0.76, 95% CI 0.68 to 0.85, p<0.001; online supplemental materials 6).

In the Grindr sample, vaccination uptake was associated with ever having taken PrEP for HIV (aOR 8.95, 95% CI 5.61 to 14.28, p<0.001) and agreeing that you were already immune to mpox (aOR 8.83, 95% CI 4.52 to 17.22, p<0.001; online supplemental materials 6). When including vaccination intention, being vaccinated for smallpox in 2022 or intending to be vaccinated if offered a vaccine was associated with agreeing that if you got a smallpox vaccine, you would be protected against mpox (aOR 3.25, 95% CI 1.97 to 5.35, p<0.001; online supplemental materials 6).

**Table 5** Associations between intending to completely stop any sexual contact and sociodemographic characteristics and motivational message, by sample

| Participant characteristics | Level | General population | | Grindr | |
|---|---|---|---|---|---|
| | | aOR (95% CI) | P value | aOR (95% CI) | P value |
| Gender | Male (including transman) | Ref | – | – | – |
| | Female (including transwoman) | 1.90 (1.48 to 2.45) | <0.001* | – | – |
| Sexual orientation | Straight or heterosexual | Ref | – | – | – |
| | Gay, lesbian, bisexual or queer | 0.91 (0.58 to 1.41) | 0.67 | – | – |
| Age | Range 18–98 years | 1.04 (1.03 to 1.05) | <0.001* | 1.01 (0.98 to 1.04) | 0.53 |
| | Quadratic term (age–mean)$^2$ | 1.000 (0.999 to 1.001) | 0.87 | 1.000 (0.998 to 1.001) | 0.71 |
| Region | Midlands (East and West) | Ref | – | Ref | – |
| | South (East, West, East of England) | 1.08 (0.76 to 1.54) | 0.67 | 1.18 (0.38 to 3.66) | 0.77 |
| | North (East, West, Yorkshire and the Humber) | 1.14 (0.79 to 1.66) | 0.49 | 1.48 (0.35 to 6.21) | 0.58 |
| | London | 1.17 (0.73 to 1.86) | 0.52 | 0.94 (0.27 to 3.25) | 0.92 |
| | Devolved nations (Scotland, Wales and Northern Ireland) | 1.26 (0.79 to 2.01) | 0.33 | 1.26 (0.31 to 5.15) | 0.74 |
| Dependent child in household | No | Ref | – | Ref | – |
| | Yes | 0.97 (0.72 to 1.29) | 0.82 | 0.37 (0.11 to 1.28) | 0.12 |
| Employment status | Not working | Ref | – | Ref | – |
| | Working | 1.22 (0.90 to 1.66) | 0.20 | 0.92 (0.39 to 2.18) | 0.85 |
| Frontline health or social care worker | No | Ref | – | Ref | – |
| | Yes | 0.97 (0.67 to 1.39) | 0.86 | 1.77 (0.67 to 4.65) | 0.25 |
| Education | GCSE/vocational/A-level/no formal qualifications | Ref | – | Ref | – |
| | Degree or higher (Bachelors, Masters, PhD) | 0.96 (0.73 to 1.27) | 0.79 | 1.87 (0.97 to 3.59) | 0.06 |
| Ethnicity | White British | Ref | – | Ref | – |
| | White other | 0.47 (0.26 to 0.84) | 0.01 | 0.84 (0.35 to 2.06) | 0.71 |
| | Black, Asian, other minoritised ethnicity | 0.85 (0.56 to 1.27) | 0.42 | 0.36 (0.15 to 0.85) | 0.02 |
| Marital status | Not partnered | Ref | – | Ref | – |
| | Partnered | 0.90 (0.65 to 1.24) | 0.50 | 1.11 (0.51 to 2.41) | 0.79 |
| Live alone | Live with someone else | Ref | – | Ref | – |
| | Live alone | 1.16 (0.77 to 1.73) | 0.48 | 1.12 (0.56 to 2.23) | 0.75 |
| Own chronic illness | None | Ref | – | Ref | – |
| | Present | 0.97 (0.72 to 1.31) | 0.85 | 1.10 (0.53 to 2.27) | 0.80 |
| Ever taken PrEP for HIV | No | – | – | Ref | – |
| | Yes | – | – | 1.36 (0.69 to 2.71) | 0.37 |
| Vaccinated for smallpox in 2022 | Not vaccinated | – | – | Ref | – |
| | Vaccinated | – | – | 0.71 (0.33 to 1.50) | 0.37 |
| Index of Multiple Deprivation | Deciles (1st=most deprived, 10th=least deprived) | 1.00 (0.96 to 1.06) | 0.85 | 1.01 (0.86 to 1.19) | 0.88 |
| Socioeconomic grade | ABC1 | Ref | – | Ref | – |
| | C2DE | 1.13 (0.86 to 1.49) | 0.39 | 4.15 (1.45 to 11.85) | 0.008 |
| Financial hardship | 4 (lowest hardship) to 13 (most hardship) | 0.85 (0.80 to 0.90) | <0.001* | 0.89 (0.74 to 1.07) | 0.21 |

Continued

**Table 5** Continued

| Participant characteristics | Level | General population | | Grindr | |
|---|---|---|---|---|---|
| | | aOR (95% CI) | P value | aOR (95% CI) | P value |
| Total no of sexual partners (male and female) in last 3 weeks | 0 | Ref | – | – | – |
| | 1 | 0.83 (0.59 to 1.18) | 0.30 | – | – |
| | 2–4 | 0.68 (0.39 to 1.18) | 0.17 | – | – |
| | 5 or more | 0.37 (0.14 to 0.96) | 0.04 | – | – |
| | Prefer not to say | 0.45 (0.32 to 0.64) | <0.001* | – | – |
| No of male sexual partners in last 3 weeks | 0 | – | – | Ref | – |
| | 1 | – | – | 0.52 (0.18 to 1.46) | 0.21 |
| | 2–4 | – | – | 0.38 (0.14 to 0.98) | 0.05 |
| | 5–9 | – | – | 0.21 (0.07 to 0.65) | 0.007 |
| | 10 or more | – | – | 0.20 (0.06 to 0.72) | 0.01 |
| | Prefer not to say | – | – | 0.30 (0.08 to 1.11) | 0.07 |
| Motivational message | Perceived risk of illness and necessity and efficacy of the response | 1.37 (0.97 to 1.93) | 0.08 | – | – |
| | Perceived risk of illness and benefits of the response | 1.07 (0.76 to 1.49) | 0.70 | – | – |
| | Perceived risk of illness and low perceived costs of response | 1.10 (0.79 to 1.55) | 0.57 | – | – |
| | Control | Ref | – | – | – |
| Motivational message | All motivational components | – | – | 1.43 (0.80 to 2.54) | 0.23 |
| | Control | – | – | Ref | – |

Variables were entered into the logistic regression model in blocks (block 1: sociodemographic variables and motivational message, block 2: psychological factors). Results for block 2, using pooled estimates are reported.
*p≤0.001.
aOR, adjusted Odds Ratio; GCSE, General Certificate of Secondary Education; PrEP, pre-exposure prophylaxis.

## DISCUSSION

We investigated mpox attitudes, beliefs and intended behaviours in a general population sample and three GBMSM samples. Samples differed by sociodemographic characteristics. This was reflected in intended behaviours, with GBMSM recruited from Grindr or Meta being more likely to intend to seek help immediately for mpox symptoms, completely stop sexual contact when symptomatic, and be vaccinated for smallpox if advised. High rates of smallpox vaccination in these samples perhaps reflect that they were mostly educated and working, and thus more likely to be health literate and engaged with services. This was also reflected in knowledge about mpox, with Grindr and Meta samples being more likely to correctly identify mpox transmission modes and the symptoms of mpox. Within the general population, rates of 'don't know' answers were high, with 18.4% selecting 'don't know' when asked what the main symptoms of mpox were, and 18.7%–24.2% selecting 'don't know' for transmission modes (except for having sex with someone who has mpox, where 13.9% selected 'don't know'). This is similar to other surveys of the general population, where 24% of people were not sure if mpox usually spreads by close contact with an infected person.[45]

There were no effects of motivational messaging on our behavioural outcomes, except for in the Meta sample for intention to self-isolate, share details of all recent sexual contacts and be vaccinated for smallpox if advised. For these outcomes, intention to engage with protective behaviours was higher in the control group. In practice, however, intentions were very similar (means changed by 0.1–0.4) and this is unlikely to represent a meaningful difference. Another study investigating gonorrhoea reinfection in young adults has also found that those in the control group (who received monthly texts reminding them to update their contact details) were less likely to be reinfected than those in the intervention group (who received a series of texts that were educational and used behaviour change techniques).[46] These messages were specifically designed to decrease stigma, and partner numbers were higher in the intervention arm. A systematic review of intervention communications during the COVID-19 pandemic found that messages about the personal and collective benefits of vaccination had mixed effects on vaccination intention, with some suggestions that this approach may be more effective in more strongly hesitant individuals.[28] These results highlight the need for empirical testing of

**Table 6** Associations between intending to completely stop any sexual contact and psychological and contextual factors, by sample

| Factor | Level | General population | | Grindr | |
|---|---|---|---|---|---|
| | | aOR (95% CI) | P value | aOR (95% CI) | P value |
| Amount heard about mpox | I have not seen or heard anything (1) to I have seen or heard a lot (3) | 1.42 (1.08 to 1.86) | 0.01 | 1.38 (0.70 to 2.72) | 0.35 |
| Worry about mpox | Not at all worried (1) to extremely worried (4) | 1.11 (0.88 to 1.40) | 0.36 | 1.51 (0.82 to 2.77) | 0.19 |
| Perceived risk of mpox to oneself | No risk at all (1) to very high risk (5) | 0.83 (0.68 to 1.02) | 0.08 | 0.93 (0.60 to 1.44) | 0.75 |
| Perceived risk of mpox to people in UK | No risk at all (1) to very high risk (5) | 1.10 (0.90 to 1.34) | 0.35 | 0.99 (0.64 to 1.53) | 0.95 |
| Perceived susceptibility and severity | Lowest (1) to highest (5) | 1.27 (1.03 to 1.56) | 0.03 | 1.18 (0.72 to 1.94) | 0.51 |
| I am already immune to mpox | Strongly disagree, disagree, neither agree nor disagree, don't know | Ref | – | Ref | – |
| | Strongly agree and agree | 0.37 (0.26 to 0.53) | <0.001* | 0.78 (0.32 to 1.90) | 0.59 |
| People who catch mpox usually make a full recovery, even if they do not receive any treatment | Strongly disagree (1) to strongly agree (5) | 0.87 (0.74 to 1.01) | 0.08 | 0.66 (0.45 to 0.97) | 0.04 |
| My personal behaviour has an impact on how mpox spreads | Strongly disagree (1) to strongly agree (5) | 1.03 (0.92 to 1.16) | 0.57 | 1.42 (1.08 to 1.87) | 0.01 |
| My life has been negatively affected by changes made in response to the mpox outbreak | Strongly disagree (1) to strongly agree (5) | 0.70 (0.62 to 0.80) | <0.001* | 0.73 (0.55 to 0.97) | 0.03 |
| The risks of mpox are being exaggerated | Strongly disagree (1) to strongly agree (5) | 0.77 (0.67 to 0.88) | <0.001* | 0.81 (0.58 to 1.11) | 0.18 |
| mpox is only a risk to men who are gay, bisexual or have sex with men | Strongly disagree (1) to strongly agree (5) | 0.87 (0.77 to 0.97) | 0.02 | 0.77 (0.58 to 1.03) | 0.08 |
| Perceived knowledge | Lowest (0) to highest (3) | 1.00 (0.88 to 1.15) | 0.94 | 0.86 (0.58 to 1.28) | 0.46 |
| Knowledge of mpox symptoms | Identified no symptoms (0) to identified four symptoms (4) | 1.14 (1.03 to 1.26) | 0.01 | 1.29 (1.00 to 1.66) | 0.05 |
| Knowledge of mpox transmission | Lowest (0) to highest (6) | 1.18 (1.08 to 1.29) | <0.001* | 1.31 (1.00 to 1.73) | 0.05 |

Variables were entered into the logistic regression model in blocks (block 1: sociodemographic variables and motivational message, block 2: psychological factors). Results for block 2, using pooled estimates are reported.
*p≤0.001.
aOR, adjusted Odds Ratio.

messages on behavioural outcomes before widespread use in public health.

Generally, higher intentions were seen in the Grindr and Meta samples, except for self-isolation. However, rates should be interpreted with caution, as intended behaviour does not always translate to enacted behaviour.[47] This was shown to be the case for intended and actual engagement with the UK contact tracing programme during the COVID-19 pandemic.[11] Rates of having already been vaccinated for smallpox in 2022 were especially high (32% Grindr, 41% Meta), again suggesting our sample may have been particularly interested in the topic of the survey, especially given issues with access to vaccination.[48] Evidence suggests that while one-off sexual encounters may make up only a minority of sexual interactions, they could account for a large proportion of mpox transmission.[49 50] Furthermore, if encounters are anonymous, this will affect contact tracing efforts. Therefore, very high rates of intending to completely stop sexual contact with others if symptomatic (91% and 93% in Grindr and Meta samples, respectively) are encouraging. In the UK, the public health agency (UK Health Security Agency) has been working in conjunction with community-based organisations and charities to raise awareness of mpox and how to prevent transmission in GBMSM as the population most affected. Higher rates of knowledge and behavioural intentions suggest that this targeted messaging has been effective in increasing knowledge and driving protective behaviours. However, efforts must be taken to ensure that messaging is not stigmatising.[51] Public health efforts in

**Table 7** Associations between intending to share details of every sexual contact in the last 7 days and sociodemographic characteristics and motivational message, by sample

| Participant characteristics | Level | General population | | Grindr | |
|---|---|---|---|---|---|
| | | B (95% CI) | P value | B (95% CI) | P value |
| Gender | Male (including transman) | Ref | – | – | – |
| | Female (including transwoman) | 0.16 (0.07 to 0.24) | <0.001* | – | – |
| Sexual orientation | Straight or heterosexual | Ref | – | – | – |
| | Gay, lesbian, bisexual or queer | 0.12 (−0.05 to 0.28) | 0.16 | – | – |
| Age | Range 18–98 years | 0.008 (0.005 to 0.011) | <0.001* | 0.003 (−0.005 to 0.011) | 0.46 |
| | Quadratic term (age–mean)$^2$ | 0.00017 (−0.00034 to −0.00001) | 0.04 | 0.0003 (−0.0002 to 0.0008) | 0.19 |
| Region | Midlands (East and West) | Ref | – | Ref | – |
| | South (East, West, East of England) | 0.04 (−0.16 to 0.08) | 0.50 | 0.08 (−0.23 to 0.39) | 0.60 |
| | North (East, West, Yorkshire and the Humber) | 0.07 (−0.19 to 0.06) | 0.29 | 0.31 (−0.05 to 0.68) | 0.09 |
| | London | 0.05 (−0.11 to 0.22) | 0.53 | 0.12 (−0.25 to 0.49) | 0.52 |
| | Devolved nations (Scotland, Wales and Northern Ireland) | 0.09 (−0.06 to 0.25) | 0.24 | 0.26 (0.21 to 0.73) | 0.27 |
| Dependent child in household | No | Ref | – | Ref | – |
| | Yes | 0.12 (−0.22 to −0.01) | 0.03 | 0.22 (−0.64 to 0.20) | 0.30 |
| Employment status | Not working | Ref | – | Ref | – |
| | Working | 0.10 (−0.20 to 0.00) | 0.05 | 0.01 (−0.26 to 0.23) | 0.90 |
| Frontline health or social care worker | No | Ref | – | Ref | – |
| | Yes | 0.04 (−0.17 to 0.09) | 0.57 | 0.08 (−0.19 to 0.34) | 0.57 |
| Education | GCSE/vocational/A-level/No formal qualifications | Ref | – | Ref | – |
| | Degree or higher (Bachelors, Masters, PhD) | 0.00 (−0.09 to 0.09) | 0.97 | 0.13 (−0.06 to 0.32) | 0.19 |
| Ethnicity | White British | Ref | – | Ref | – |
| | White other | 0.29 (−0.51 to −0.08) | 0.008 | 0.00 (−0.25 to 0.25) | 0.98 |
| | Black, Asian, other minoritised ethnicity | 0.17 (−0.32 to −0.01) | 0.03 | 0.01 (−0.29 to 0.30) | 0.97 |
| Marital status | Not partnered | Ref | – | Ref | – |
| | Partnered | 0.03 (−0.08 to 0.15) | 0.55 | 0.10 (−0.11 to 0.32) | 0.35 |
| Live alone | Live with someone else | Ref | – | Ref | – |
| | Live alone | 0.04 (−0.17 to 0.08) | 0.50 | 0.21 (0.01 to 0.40) | 0.04 |
| Own chronic illness | None | Ref | – | Ref | – |
| | Present | 0.04 (−0.14 to 0.05) | 0.40 | 0.31 (0.11 to 0.51) | 0.003 |
| Ever taken PrEP for HIV | No | – | – | Ref | – |
| | Yes | – | – | 0.09 (−0.10 to 0.28) | 0.35 |
| Vaccinated for smallpox in 2022 | Not vaccinated | – | – | Ref | – |
| | Vaccinated | – | – | 0.07 (−0.28 to 0.14) | 0.50 |
| Index of Multiple Deprivation | Deciles (1st=most deprived, 10th=least deprived) | 0.00 (−0.02 to 0.02) | 0.87 | 0.03 (−0.02 to 0.08) | 0.19 |
| Socioeconomic grade | ABC1 | Ref | – | Ref | – |
| | C2DE | 0.12 (−0.21 to −0.03) | 0.01 | 0.04 (−0.22 to 0.29) | 0.78 |
| Financial hardship | 4 (lowest hardship) to 13 (most hardship) | 0.05 (−0.07 to −0.03) | <0.001* | 0.02 (−0.07 to 0.04) | 0.53 |

Continued

**Table 7** Continued

| Participant characteristics | Level | General population | | Grindr | |
|---|---|---|---|---|---|
| | | B (95% CI) | P value | B (95% CI) | P value |
| Total no of sexual partners (male and female) in last 3 weeks | 0 | Ref | – | – | – |
| | 1 | 0.06 (−0.05 to 0.17) | 0.28 | – | – |
| | 2–4 | 0.07 (−0.14 to 0.27) | 0.53 | – | – |
| | 5 or more | 0.09 (−0.45 to 0.28) | 0.64 | – | – |
| | Prefer not to say | 0.37 (−0.49 to −0.24) | <0.001* | – | – |
| No of male sexual partners in last 3 weeks | 0 | – | – | Ref | – |
| | 1 | – | – | 0.02 (−0.27 to 0.22) | 0.84 |
| | 2–4 | – | – | 0.02 (−0.24 to 0.21) | 0.90 |
| | 5–9 | – | – | 0.08 (−0.38 to 0.23) | 0.62 |
| | 10 or more | – | – | 0.58 (−0.96 to −0.20) | 0.003 |
| | Prefer not to say | – | – | 0.09 (−0.52 to 0.34) | 0.67 |
| Motivational message | Perceived risk of illness and necessity and efficacy of the response | 0.03 (−0.15 to 0.08) | 0.55 | – | – |
| | Perceived risk of illness and benefits of the response | 0.03 (−0.08 to 0.14) | 0.59 | – | – |
| | Perceived risk of illness and low perceived costs of response | 0.01 (−0.12 to 0.10) | 0.86 | – | – |
| | Control | Ref | – | – | – |
| Motivational message | All motivational components | – | – | 0.06 (−0.23 to 0.10) | 0.45 |
| | Control | – | – | Ref | – |

A higher score indicates greater intention to share details. Variables were entered into the linear regression model in blocks (block 1: sociodemographic variables and motivational message, block 2: psychological factors). Results for block 2, using pooled estimates are reported.
*p≤0.001.
GCSE, General Certificate of Secondary Education; PrEP, pre-exposure prophylaxis.

future outbreaks should use a similar model to increase knowledge about transmission and engagement with protective behaviours in affected populations.[31]

It is notable that 19% (Grindr, Meta) to 30% of people (general population, Savanta GBMSM) agreed that it is best to avoid physical contact with GBMSM because of the mpox outbreak, perhaps also reflecting a change in behaviour in this population. Recent decreases in some STIs in GBMSM, and the decline in mpox transmission in England, suggest this may be the case.[6] A study conducted in the USA of men who have sex with men and transgender women found that 56% of participants reported changing their sexual behaviour due to the mpox outbreak, with most participants limiting the number of their sexual partners; 25% of the sample became abstinent or avoided having any type of sex.[52]

Within the general population, women, people who were older and those with lower financial hardship were more likely to intend to carry out protective behaviours. This pattern was also widely seen during the COVID-19 pandemic[11 53 54] and previous outbreaks.[55] This highlights the importance of considering health equity issues to ensure effective outbreak control and has implications for policy-makers. Offering financial support for protective behaviours, especially those that may affect people's ability to earn an income such as self-isolation, is likely to increase engagement, especially for those in lower income settings. Few sociodemographic characteristics were associated with outcomes in the Grindr sample. This could be because the Grindr sample differed from the general population sample on key characteristics (more educated, higher socioeconomic grade, less financial hardship), or be a function of the way analyses were conducted with all variables being entered into a single regression model. The Grindr sample, being smaller, also had about half the statistical power of the general population sample.

Intention to engage in protective behaviours in the general population was also associated with psychological factors such as greater worry about mpox, perceived risk of mpox to others (but there was little evidence of an association with perceived risk to oneself), perceived susceptibility and severity of mpox, and greater knowledge about transmission. This pattern of results was also seen during the COVID-19 and influenza A H1N1 pandemics.[12 53 56–58] COVID-19 vaccination intention was also associated with

**Table 8** Associations between intending to share details of every sexual contact in the last 7 days and psychological and contextual factors, by sample

| Factor | Level | General population | | Grindr | |
|---|---|---|---|---|---|
| | | B (95% CI) | P value | B (95% CI) | P value |
| Amount heard about mpox | I have not seen or heard anything (1) to I have seen or heard a lot (3) | 0.04 (−0.13 to 0.05) | 0.43 | 0.01 (−0.18 to 0.20) | 0.92 |
| Worry about mpox | Not at all worried (1) to extremely worried (4) | 0.08 (0.00 to 0.16) | 0.06 | 0.19 (0.01 to 0.36) | 0.03 |
| Perceived risk of mpox to oneself | No risk at all (1) to very high risk (5) | 0.04 (−0.11 to 0.03) | 0.27 | 0.06 (−0.18 to 0.06) | 0.34 |
| Perceived risk of mpox to people in UK | No risk at all (1) to very high risk (5) | 0.07 (0.00 to 0.14) | 0.06 | 0.01 (−0.14 to 0.12) | 0.85 |
| Perceived susceptibility and severity | Lowest (1) to highest (5) | 0.12 (0.05 to 0.19) | <0.001* | 0.09 (−0.24 to 0.06) | 0.24 |
| I am already immune to mpox | Strongly disagree, disagree, neither agree nor disagree, don't know | Ref | – | Ref | – |
| | Strongly agree and agree | 0.18 (−0.32 to −0.04) | 0.01 | 0.09 (−0.37 to 0.20) | 0.55 |
| People who catch mpox usually make a full recovery, even if they do not receive any treatment | Strongly disagree (1) to strongly agree (5) | 0.01 (−0.06 to 0.04) | 0.71 | 0.06 (−0.17 to 0.04) | 0.23 |
| My personal behaviour has an impact on how mpox spreads | Strongly disagree (1) to strongly agree (5) | 0.04 (0.01 to 0.08) | 0.03 | 0.02 (−0.06 to 0.10) | 0.64 |
| My life has been negatively affected by changes made in response to the mpox outbreak | Strongly disagree (1) to strongly agree (5) | 0.10 (−0.15 to −0.06) | <0.001* | 0.03 (−0.11 to 0.06) | 0.56 |
| The risks of mpox are being exaggerated | Strongly disagree (1) to strongly agree (5) | 0.11 (−0.15 to −0.06) | <0.001* | 0.15 (−0.25 to −0.05) | 0.004 |
| Mpox is only a risk to men who are gay, bisexual or have sex with men | Strongly disagree (1) to strongly agree (5) | 0.02 (−0.06 to 0.02) | 0.29 | 0.07 (−0.16 to 0.01) | 0.10 |
| Perceived knowledge | Lowest (0) to highest (3) | 0.05 (0.00 to 0.09) | 0.04 | 0.02 (−0.10 to 0.13) | 0.78 |
| Knowledge of mpox symptoms | Identified no symptoms (0) to identified four symptoms (4) | 0.05 (0.02 to 0.08) | 0.004 | 0.05 (−0.02 to 0.13) | 0.14 |
| Knowledge of mpox transmission | Lowest (0) to highest (6) | 0.05 (0.02 to 0.08) | <0.001 | 0.05 (−0.03 to 0.13) | 0.22 |

A higher score indicates greater intention to share details. Variables were entered into the linear regression model in blocks (block 1: sociodemographic variables and motivational message, block 2: psychological factors). Results for block 2, using pooled estimates are reported.
*p≤0.001.

perceived risk to others, but not to oneself.[36] Few associations were significant in the Grindr sample. In addition to the decreased statistical power, this may also reflect the different context and experience that participants in this group had. While for our general population sample the questions about mpox risk were somewhat hypothetical, the Grindr sample was more likely to be impacted by mpox (as illustrated by the high vaccination rates). Factors relating to their personal experience of the outbreak may have superseded any effect of the psychological variables that we assessed. Taken together, these results suggest that clear communications about the level of risk of infection may encourage people to enact protective behaviours when appropriate.

Agreeing that the risks of mpox were being exaggerated was associated with lower intention to engage with protective behaviours. This was also the case in previous pandemics.[53 59] Other psychological factors associated with decreased intention to engage with protective behaviours were believing that you were already immune to mpox and that your life had been negatively affected by changes made in response to mpox. Attending an important event was one of the reasons given for breaking self-isolation during the aH1N1 pandemic.[60] The peak of the mpox outbreak in the UK occurred in July 2022, a period coinciding with the summer holiday and festival season, including Pride events. It is important to be conscious of asking people to self-isolate or quarantine over periods encompassing large public events that may be happening.

Behaviour-specific beliefs were also associated with intentions. For self-isolation, greater intention was associated with greater social norms (thinking that others would also self-isolate). Greater social norms were associated with fewer outings during lockdown[14] and increased vaccination uptake and intention (in oneself and one's child)[15 61] during the COVID-19 pandemic. Social norms were also associated with enacting protective behaviours, including self-isolation, during previous outbreaks.[55 62] Lower self-isolation intention was associated with thinking that self-isolation would have a negative impact on your work. In this study, as in previous research,[11] the main reasons for not being able to self-isolate were needing to go out for essentials (food/medicines), for a walk or some other exercise and for work. Results suggest that communications emphasising that others are also engaging with the protective behaviour may improve engagement with self-isolation. Research investigating the relative importance of different psychological factors and specific needs in behavioural decisions could contribute to the design of more effective communications.

In the UK, people with suspected mpox were directed to call sexual health services. Some people may have found this stigmatising as mpox is not a STI.[63] Stigma surrounding having other STIs is associated with not seeking help.[64] In our study, immediate help seeking was associated with being willing to contact a sexual health clinic. These findings suggest that widening

recommended points of contact with health services to include non-stigmatising routes for suspected mpox cases or contacts may be beneficial and could have increased help seeking.

In the Grindr sample, having been vaccinated for smallpox in 2022 was associated with having ever taken PrEP for HIV. This is probably due to how the vaccine was rolled out in the UK. GBMSM identified by sexual health services as being at highest risk of exposure—using markers similar to those used to assess eligibility for PrEP—were invited to be vaccinated.[65] Being vaccinated was also strongly associated with thinking that you were already immune to mpox in this survey. As this is a cross-sectional survey, we cannot infer the direction of results, but it is likely that respondents believed they were immune to mpox because of their recent vaccination. This is interesting given limited information about the effectiveness of the vaccine at that time.

Strengths of the study include the collection of data from four large samples, including the general population and the population most affected by the recent mpox outbreak (GBMSM). We would also caution readers about several caveats for this study, including that participant sociodemographic characteristics differed by the recruitment method. Participants recruited through Grindr and Meta were more likely to be highly educated, higher socioeconomic grade and have less financial hardship. We are unsure whether this is representative of the sociodemographic characteristics of active Grindr and Meta users as we were unable to find, access or interpret published statistics of the sociodemographic profile of users.[66] Where published, most statistics outline only age and gender, rather than the additional characteristics we were interested in.[67] High rates of smallpox vaccination in these samples suggest that these groups are likely to be interested in and have personal experience relating to the outbreak and may have been more likely to engage with protective behaviours. We do not know if survey respondents have representative beliefs, knowledge and intended behaviours with reference to the general population, GBMSM generally and GBMSM who use Grindr and Meta. While overall rates of beliefs, knowledge and intended behaviours may be affected by sampling and should be taken with caution, associations within the data are likely to hold true.[68]

Other limitations relate to the survey and statistical measures used. The use of a cross-sectional survey means that answers may have been influenced by social desirability and recall bias, although the anonymity of a written survey may have mitigated these effects to some degree. Rates of intended behaviour are often higher than enacted behaviours.[11 47] Therefore, rates of intended behaviours should be taken with caution. Our motivational messages were very brief and only repeated once, before measuring intended behaviours and behaviour-specific attitudes. A fully developed public communications campaign where people are exposed to coproduced and repeated messages may have more influence. There

were high rates of actual and intended vaccination in the Grindr sample (94%). This may have affected our ability to detect associations. Variables were entered into regression models in blocks, with results reported for the final block (essentially all variables entered together). Therefore, we investigated the independent effect of a variable, accounting for all other variables included.

This study investigated mpox beliefs, knowledge and intended behaviours in a general population sample and in GBMSM (those most affected by the 2022 outbreak). Intended uptake of protective behaviours differed by behaviour. GBMSM generally had higher intention to engage with protective behaviours, apart from self-isolation. This may have been a function of sampling. Higher knowledge about mpox symptoms and transmission in GBMSM samples suggests that the public health messaging carried out by multiple stakeholders including the UK Health Security Agency, charities and via grass-roots community efforts has been successful and a similar model should be used in future outbreaks. There was no impact of additional motivational messaging on intended uptake of protective behaviours. Associations between increased financial hardship and lower intention to enact protective behaviours suggest that providing financial support to those affected in future outbreaks may increase uptake.

**Author affiliations**
[1]Department of Psychological Medicine, King's College London, London, UK
[2]Institute of Health Informatics, University College London, London, UK
[3]Norwich Medical School, University of East Anglia, Norwich, UK
[4]School of Psychological Science, University of Bristol, Bristol, UK
[5]UK Health Security Agency, London, UK

**Acknowledgements** We would like to thank the Terrence Higgins Trust for their support with participant recruitment, and Savanta for hosting the survey. We would also like to thank study participants, for taking their time to complete the survey.

**Contributors** LES, HP, JB, TM, IO, RA, LY and GJR conceptualised the study and contributed to survey materials. LS completed analyses with guidance from HP, JB and GJR. LES wrote the first draft of the manuscript. LES, HP, JB, TM, IO, RA, LY and GJR contributed to, and approved, the final manuscript. LES is guarantor. The corresponding author attested that all listed authors meet authorship criteria and that no others meeting the criteria have been omitted.

**Funding** LES, JB, RA and GJR are supported by the National Institute for Health and Care Research Health Protection Research Unit (NIHR HPRU) in Emergency Preparedness and Response, a partnership between the UK Health Security Agency, King's College London and the University of East Anglia. Additionally, IO, LY, TM and RA are funded by the NIHR HPRU in Behavioural Science and Evaluation, a partnership between UKHSA and the University of Bristol. The study was funded by NIHR. For the purpose of open access, the author has applied a Creative Commons Attribution (CC BY) licence to any Author Accepted Manuscript version arising.

**Disclaimer** The views expressed are those of the authors and not necessarily those of the NIHR, UKHSA, or the Department of Health and Social Care.

**Competing interests** HP has received additional salary support from Public Health England and NHS England and receives funding from the National Institute of Health Research. HP receives consultancy fees to his employer from Ipsos MORI and has a PhD student who works at and has fees paid by AstraZeneca. IO and RA are employees of UKHSA.

**Patient and public involvement** Patients and/or the public were involved in the design, or conduct, or reporting, or dissemination plans of this research. Refer to the Methods section for further details.

**Patient consent for publication** Not applicable.

**Ethics approval** Ethical approval for this study was given by the King's College London Psychiatry, Nursing and Midwifery Research Ethics Panel (reference number: LRS/DP-21/22-32287). Participants gave informed written consent before beginning survey materials.

**Provenance and peer review** Not commissioned; externally peer reviewed.

**Data availability statement** Data are available in a public, open access repository. Anonymised data are available online.

**ORCID iDs**
Louise E Smith http://orcid.org/0000-0002-1277-2564
Henry WW Potts http://orcid.org/0000-0002-6200-8804
Julii Brainard http://orcid.org/0000-0002-5272-7995
Tom May http://orcid.org/0000-0003-3077-523X
Isabel Oliver http://orcid.org/0000-0002-6106-1734
Richard Amlôt http://orcid.org/0000-0003-3481-6588
Lucy Yardley http://orcid.org/0000-0002-3853-883X

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
