## [Reviewer comments · BMJ Open]

ARTICLE DETAILS

TITLE (PROVISIONAL)	Did mpox knowledge, attitudes, and beliefs affect intended behaviour in the general population and men who are gay, bisexual, and who have sex with men? An online cross-sectional survey in the UK
AUTHORS	Smith, Louise; Potts, Henry; Brainard, Julii; May, Tom; Oliver, Isabel; Amlôt, Richard; Yardley, Lucy; Rubin, G James

VERSION 1 – REVIEW

REVIEWER	Christopher Owens Texas A&M University, Department of Health Behavior
REVIEW RETURNED	11-Jan-2023

GENERAL COMMENTS	My comments are bellow for bmjopen-2022-070882 titled, "Mpox knowledge, attitudes, beliefs, and intended behaviour in the general population and men who are gay, bisexual, and who have sex with men."  1. In the Introduction, I would like the authors to briefly provide a rationale for why there might be differences in mpox knowledge, attitudes, and behaviours between the general population and GBMSM population. 2. In the Methods, were the validated scales also theory-based, such as informed by the PMT (or other common theories regarding this topic such as the HBM)? 3. In the Supplemental Materials 1, I think it would be beneficial if the authors can have a heading called messages or something equivalent. Because of the length of the survey items, I missed that the messaging prompts are in the intended behaviour header. 4. The authors conflated sex and gender, and I would like the transgender group to be separated from those who are cisgender. For example, the authors could include transmen and transwomen into the non-binary or gender minority group. 5. The Discussion was lacking in terms of research, practice/clinical, and policy implications. For example, what do the authors suggest then for mpox (or infectious disease) messaging? 5.
--

REVIEWER	Abdullahm Asiri Ministry of Health Riyadh
REVIEW RETURNED	24-Feb-2023

GENERAL COMMENTS	I wonder if the number of responses you got from Grindr and Meta corresponds to the estimated census of GBMSM in the study population. otherwise, the study is well designed, well executed and well
---

	interpreted.
REVIEWER	Muhammad Mainuddin Patwary Environment and Sustainability Research Initiative
REVIEW RETURNED	02-Apr-2023

GENERAL COMMENTS	Thank you for the opportunity to review this manuscript. I read the manuscript on Mpox knowledge, attitudes, beliefs, and intended behavior in the general population and men who are gay, bisexual, and who have sex with men with much care. I found the study to be very interesting and informative, and I appreciate the effort you put into conducting this research. I particularly liked how you used an experimental approach to investigate the impact of different brief communication approaches on intended behavior. This is an innovative way to examine how people respond to health messages and could have important implications for developing effective health interventions. However, I also noticed some areas where the study could be improved. For example, while your sample size was large, it was recruited from a market research company, a dating app (Grindr), and targeted adverts on Meta (Facebook and Instagram). This may not be representative of the general population or GBMSM community. How, you tackle this? Additionally, while you used logistic regression analyses to examine factors associated with intention to share details of every sexual contact in both general population and GBMSM samples, it would be interesting to see more complex relationships between variables such as structural equation modeling (SEM), path analysis, or multilevel modeling. These types of analyses can help to examine more complex relationships between variables and can be useful when there are multiple predictors or outcomes of interest. Another concern, please provide sources or validation of every measures used in this study. Overall, I think your study makes an important contribution to our understanding of Mpox knowledge, attitudes, beliefs, and intended behavior in these populations. Thank you for your hard work on this project.
--

VERSION 1 – AUTHOR RESPONSE

Reviewer: 1

Dr. Christopher Owens, Texas A&M University Comments to the Author:

My comments are bellow for bmjopen-2022-070882 titled, "Mpox knowledge, attitudes, beliefs, and intended behaviour in the general population and men who are gay, bisexual, and who have sex with men."

1. In the Introduction, I would like the authors to briefly provide a rationale for why there might be differences in mpox knowledge, attitudes, and behaviours between the general population and GBMSM population.

We have added that previous efforts to decrease HIV incidence, and current targeted efforts by public health agencies, community-based organisations, charities, the dating app Grindr, and organisers of Pride events to communicate accurate scientific information about mpox to GBMSM are likely to have resulted in increased knowledge about safe sex practices and increased knowledge about mpox in GBMSM compared to the general population (p7, Introduction). Beliefs about mpox, e.g. perceived

risk and worry, were also likely to be different in GBMSM compared to the general population due to the outbreak mostly affecting this group.

2. In the Methods, were the validated scales also theory-based, such as informed by the PMT (or other common theories regarding this topic such as the HBM)?

Validated scales used in the study were theory-based. Items from Flu Telephone Survey Template (FluTEST, DOI: 10.3310/hsdr02410) were based on the PMT as they measure likelihood to engage in protective behaviours. Items in the Perceptions About Hazardous Substances (PATHS) questionnaire were based on theories and models of risk perception (e.g. DOI: 10.1007/s12529-008-9002-8, DOI: 10.1126/science.3563507).

3. In the Supplemental Materials 1, I think it would be beneficial if the authors can have a heading called messages or something equivalent. Because of the length of the survey items, I missed that the messaging prompts are in the intended behaviour header.

The header “motivational messaging” has been added to Supplementary Materials 1 (p13).

4. The authors conflated sex and gender, and I would like the transgender group to be separated from those who are cisgender. For example, the authors could include transmen and transwomen into the non-binary or gender minority group.

Participants were asked if their gender was the same as the gender they were assigned at birth. We have added a row to table 1 and reported ns for people who answered “no” or “prefer not to say”. Due to small cell counts and the potential to identify individuals, we have not separated these categories out in the analyses.

5. The Discussion was lacking in terms of research, practice/clinical, and policy implications. For example, what do the authors suggest then for mpox (or infectious disease) messaging?

We have added research, practice, and policy implications throughout the discussion (p28 to p32).

This includes highlighting the need for empirical testing of messaging on behaviours before widespread rollout, engagement with community-based organisations and charities in future outbreaks to increase knowledge and uptake of protective behaviours in affected populations, and financial reimbursement for those required to self-isolate among others.

Reviewer: 2

Dr. Abdullahm Asiri, Ministry of Health Riyadh Comments to the Author:

I wonder if the number of responses you got from Grindr and Meta corresponds to the estimated census of GBMSM in the study population.

Official estimates from the Office of National Statistics for the UK in 2020, indicate that 93.6% identify as straight or heterosexual. This is similar to the 92.7% reported in our general population sample.

We would not expect the number of responses from Grindr and Meta to be representative of the census as people using Grindr and Meta may have specific motivations to use the apps / sites (e.g. to find a partner) and are likely those who feel more comfortable using the internet (e.g. users of Meta are not evenly distributed by age and gender [<https://www.statista.com/statistics/1315413/uk-meta-audiences-by-age-and-gender/>]). We tried to investigate whether respondents from the study were representative of Grindr and Meta users, but were unable to as we could not find, access, or interpret published statistics of the sociodemographic profile of users (i.e. “11% of dating service users in the UK use Grindr”, but what is the profile of “dating service users”?

[<https://www.statista.com/forecasts/1335156/grindr-dating-brand-profile-in-the-uk>]). Furthermore, where statistics are published, they tend only to outline information about age and gender, rather than more detailed socio-demographic information about e.g. socio-economic status, household income, and education. A statement to this effect has been added to the manuscript limitations.

In addition to this, people constantly download and delete apps, so that the number and profile of users changes constantly (research suggests that retention rate of “social” apps after 30 days of

installation is 2.8% [<https://www.statista.com/statistics/259329/ios-and-android-app-user-retention-rate>]).

Otherwise, the study is well designed, well executed and well interpreted. We thank the reviewer for their kind comments.

Reviewer: 3

Mr. Muhammad Mainuddin Patwary , Environment and Sustainability Research Initiative Comments to the Author:

Thank you for the opportunity to review this manuscript. I read the manuscript on Mpox knowledge, attitudes, beliefs, and intended behavior in the general population and men who are gay, bisexual, and who have sex with men with much care. I found the study to be very interesting and informative, and I appreciate the effort you put into conducting this research.

I particularly liked how you used an experimental approach to investigate the impact of different brief communication approaches on intended behavior. This is an innovative way to examine how people respond to health messages and could have important implications for developing effective health interventions.

We thank the reviewer for their kind comments.

However, I also noticed some areas where the study could be improved. For example, while your sample size was large, it was recruited from a market research company, a dating app (Grindr), and targeted adverts on Meta (Facebook and Instagram). This may not be representative of the general population or GBMSM community. How, you tackle this?.

We acknowledge that the beliefs, knowledge and intended behaviours of the samples reported in the study may not be representative of those of the general population, GBMSM generally, and GBMSM who use Grindr and Meta (p31, Discussion). Therefore, readers should take percentages of beliefs, knowledge and intended behaviours with caution. A note to this effect has been added to the manuscript. However, according to Kohler 2019 (ref 65), associations within the data are likely to remain valid, so effect sizes reported in analyses of associations are likely to be representative of the wider populations.

Additionally, while you used logistic regression analyses to examine factors associated with intention to share details of every sexual contact in both general population and GBMSM samples, it would be interesting to see more complex relationships between variables such as structural equation modeling (SEM), path analysis, or multilevel modeling. These types of analyses can help to examine more complex relationships between variables and can be useful when there are multiple predictors or outcomes of interest.

We did not have prior hypotheses relating to more complex relationships between variables, as would require structural equation modelling, path analysis, or multilevel modelling, so we do not think it would be appropriate to add these in now.

Another concern, please provide sources or validation of every measures used in this study.

As much as possible, we used validated measures and survey items previously used in research for this survey (refs 11, 14, 33-36, p8, Study Materials section of Methods). Socio-demographic characteristics were measured using items from the Office for National Statistics, UK Health Security Agency, Ipsos MORI, Organisation for Economic Co-operation and Development, and the UK Government (refs 38 to 42, p12, Socio-demographic characteristics section of Methods).

All survey materials underwent public involvement from four lay individuals (p12, Patient and public involvement section of Methods), and any problems with understanding or clarity were addressed before commencing data collection.

Overall, I think your study makes an important contribution to our understanding of Mpox knowledge, attitudes, beliefs, and intended behavior in these populations. Thank you for your hard work on this project.

We thank the reviewer for their kind comments.

VERSION 2 – REVIEW

REVIEWER	Christopher Owens Texas A&M University, Department of Health Behavior
REVIEW RETURNED	24-May-2023

GENERAL COMMENTS	I have minor comments. I think the authors addressed my prior concerns.  1. Title. I would include the UK in the title, such as "...men in the UK?" or "...survey in the UK" 2. Introduction. Update the statistics of mpox cases and death 3. Introduction and Discussion. I would add the following article as it does discuss mitigation strategies MSM in the US reported. Hubach RD, Owens C. Findings on the monkeypox exposure mitigation strategies employed by men who have sex with men and transgender women in the United States. Archives of sexual behavior. 2022 Nov;51(8):3653-8. Additionally, I would add the following article as it shows MSM in the US are concerned about messaging and media portrayals of mpox. Owens C, Hubach RD. An Exploratory Study of the Mpox Media Consumption, Attitudes, and Preferences of Sexual and Gender Minority People Assigned Male at Birth in the United States. LGBT health. 2023 Feb 3. 4. Introduction. I am still missing what information, literature, etc. gaps this manuscript fills. How is the study innovative and significant? What implications (research, clinical, intervention, policy) can results from this study provide? I believe the authors noted it in the Discussion, but I would like more explicitness in the Introductions. 5. Thank you for the clarification on how sex and gender were measured. I still think the authors should have created 3 categories for gender: cisgender male (male gender with yes gender is the same as birth), cisgender female (female gender with yes gender is the same as birth), and transgender and gender diverse (all others). I think it would be beneficial for the authors when they describe their gender categories to briefly (in 1 sentence) describe why they coded gender as they did. It feels odd to me because trans men are typically excluded from the GBMSM group.
---

REVIEWER	Muhammad Mainuddin Patwary Environment and Sustainability Research Initiative
REVIEW RETURNED	16-May-2023

GENERAL COMMENTS	Thank you author for addressing my comments. The manuscript can be accepted without further concerns.
---

VERSION 2 – AUTHOR RESPONSE

Reviewer: 3

Mr. Muhammad Mainuddin Patwary , Environment and Sustainability Research Initiative

Comments to the Author:

Thank you author for addressing my comments. The manuscript can be accepted without further concerns.

We thank the reviewer for their kind comments.

Reviewer: 1

Dr. Christopher Owens, Texas A&M University

Comments to the Author:

I have minor comments. I think the authors addressed my prior concerns.

We thank the reviewer for their time and comments.

1. Title. I would include the UK in the title, such as "...men in the UK?" or "...survey in the UK"

"In the UK" has been added to the title.

2. Introduction. Update the statistics of mpox cases and death

Mpox statistics have been updated for August 2023 (first paragraph of Introduction).

3. Introduction and Discussion. I would add the following article as it does discuss mitigation strategies MSM in the US reported. Hubach RD, Owens C. Findings on the monkeypox exposure mitigation strategies employed by men who have sex with men and transgender women in the United States. Archives of sexual behavior. 2022 Nov;51(8):3653-8. Additionally, I would add the following article as it shows MSM in the US are concerned about messaging and media portrayals of mpox. Owens C, Hubach RD. An Exploratory Study of the Mpox Media Consumption, Attitudes, and Preferences of Sexual and Gender Minority People Assigned Male at Birth in the United States. LGBT health. 2023 Feb 3.

We thank the reviewer for drawing our attention to these articles. They are now included in the manuscript as reference #52 and reference #51 respectively.

4. Introduction. I am still missing what information, literature, etc. gaps this manuscript fills. How is the study innovative and significant? What implications (research, clinical, intervention, policy) can results from this study provide? I believe the authors noted it in the Discussion, but I would like more explicitness in the Introductions.

To the best of our knowledge, there is only one other quantitative study investigating mpox knowledge, attitudes, and beliefs in the UK. This study did not investigate the breadth of behaviours that would take people through the mpox test, trace, and isolate journey. Furthermore, questions used to measure behaviours (e.g. self isolation "I agree that people should isolate for 21 days when to avoid passing monkeypox on to others", response options: yes, no, don't know/ not sure) were framed relating to "people" rather than their own individual intended behaviour. This is likely to give different results.

In our study, we investigated the effectiveness of messages that could be used in official communications. It is important to test the effectiveness of messages before using them in official communications so as to avoid unintended negative consequences. We have investigated associations between intended behaviours and socio-demographic characteristics; this helps to form the start of an evidence base for policy and communications. We also investigated psychological characteristics that could be potentially modified by communications with the aim of increasing uptake of protective behaviours. These points have been added to the start of the last paragraph of the introduction. Additional strengths of our study are the inclusion of the general population and GBMSM sample – as mpox risk existed in both communities, but at different levels – and that the study team include members of the UK Health Security Agency, meaning that findings were operationalised in a timely manner.

5. Thank you for the clarification on how sex and gender were measured. I still think the authors should have created 3 categories for gender: cisgender male (male gender with yes gender is the same as birth), cisgender female (female gender with yes gender is the same as birth), and transgender and gender diverse (all others). I think it would be beneficial for the authors when they describe their gender categories to briefly (in 1 sentence) describe why they coded gender as they did. It feels odd to me because trans men are typically excluded from the GBMSM group.

We coded gender based on recently updated categorisations used by the Genitourinary Medicine Clinic Activity Dataset (GUMCAD) sexually transmitted infections (STI) surveillance system in England (<https://www.gov.uk/guidance/gumcad-sti-surveillance-system>, reference #38 in manuscript). This has been added to the socio-demographic characteristics section of the methods.

VERSION 3 – REVIEW

REVIEWER	Christopher Owens Texas A&M University, Department of Health Behavior
REVIEW RETURNED	09-Sep-2023
GENERAL COMMENTS	The authors addressed my comments. I have no other comments based on my previous review and the authors' recent revisions.